# Somatic NGS Analysis of DNA Damage Response (DDR) Genes *ATM*, *MRE11A*, *RAD50*, *NBN*, and *ATR* in Locally Advanced Rectal Cancer Treated with Neoadjuvant Chemo-Radiotherapy

**DOI:** 10.3390/biomedicines10123247

**Published:** 2022-12-13

**Authors:** Andrea Montori, Aldo Germani, Mario Ferri, Annalisa Milano, Teresa Valentina Ranalli, Maria Piane, Emanuela Pilozzi

**Affiliations:** 1Department of Clinical and Molecular Medicine, “Sapienza” University of Rome, Viale Regina Elena 324, 00161 Rome, Italy; 2Unit of Pathologic Morphological and Molecular Anatomy, Sant’Andrea University Hospital, Via di Grottarossa 1035, 00189 Rome, Italy; 3Department of Medical-Surgical Sciences and Translational Medicine, “Sapienza” University of Rome, Piazzale Aldo Moro 5, 00189 Rome, Italy; 4Unit of Gastrointestinal Surgery, Sant’Andrea University Hospital, Via di Grottarossa 1035, 00189 Rome, Italy; 5Unit of Oncology, Sant’Andrea University Hospital, Via di Grottarossa 1035, 00189 Rome, Italy; 6Unit of Pathology, Belcolle Hospital, Str. Sammartinese, 01100 Viterbo, Italy; 7Unit of Medical Genetics and Advanced Cellular Diagnostic, Sant’Andrea University Hospital, Via di Grottarossa 1035, 00189 Rome, Italy

**Keywords:** rectal cancer, ATM, RAD 50, MRN complex, NGS analysis

## Abstract

Background: Neoadjuvant chemo-radiotherapy (nCRT) represents the standard of care for locally advanced rectal cancer (LARC); however, there exists no biomarker that can predict the cancer’s response to treatment as less than 20% of patients experience pathological complete response (pCR). Ionizing radiations induce double strand breaks (DSBs) and trigger a DNA damage response (DDR) involving ATM, ATR, and the MRN complex (MRE11, Rad50, and NBS1). In this study, we performed an extensive mutational analysis of the genes involved in the DDR pathway in LARC patients who have undergone nCRT. Methods: 13 LARC patients with pCR and 11 LARC patients with partial response (pPR) were investigated using a NGS dedicated panel, designed for formalin-fixed paraffin-embedded (FFPE) samples, containing *ATR*, *ATM*, and *MRE11-RAD50-NBN* genes. The identified variants were classified according to guidelines’ recommendations. Results: Eight non-benign variants, six of which were observed in 3 (23%) out of 13 pCR patients, were identified. In particular, a pCR patient carried out a pathogenetic frameshift mutation in exon 21 of the *RAD50* gene. The two remaining non-benign missense variants were found in 2 (18%) out of 11 patients in the pPR group. Conclusions: Our data show that the genes involved in the Homologous Recombination (HR) pathway are rarely mutated in LARC; however, given the identification of a missense mutation in RAD 50 in one case of pCR, it could be worth exploring its potential role as a biomarker in larger series.

## 1. Introduction

Rectal cancer accounts for approximately one-third of colorectal cancer (CRC) cases but more than half of CRC-related deaths [1]. Although rectal cancers are histologically similar to colon cancers, considerable biological and clinical evidence indicates that rectal cancer must be considered as a distinct entity [2].

Over the past few decades, the standard of care for locally advanced rectal cancer (LARC)—T3–T4 and/or node positive—has evolved remarkably. An adjuvant regimen of preoperative neoadjuvant chemoradiotherapy (nCRT), followed by surgical total mesorectal excision (TME) in an 8–10 weeks lapse time represents the standard of care that has been proven to reduce the exceptionally high rate of local and distant recurrences. Despite these improvements, ~30% of patients still develop distant metastasis or local recurrence, which remains the leading cause of rectal cancer-related death [3,4].

nCRT treatment causes tumor regression that can be assessed either through nuclear magnetic resonance (NMR) and histopathological evaluation of surgical samples using a tumor regression grading (TRG) system which according to the American Joint Committee on Cancer (AJCC) is classified in four-point grades based on the absence of residual cancer cells (TRG0) or a variable residual volume of primary tumor cells (TRG1 to 3). Despite the same regimen of nCRT, tumor regression may vary since approximately 20% of patients reach a complete pathological response (pCR), i.e., the absence of viable cancer cells in the rectal wall and in the lymph nodes, whilst the remaining patients experience the absence of or a partial pathological response (pPR) [5,6,7]. In the absence of a residual tumor, these patients could potentially be spared the morbidity of rectal resection, but there is still no effective way of predicting the patients’ response to nCRT. Many studies have sought biomarkers that would predict the patients’ response to radiotherapy or chemoradiotherapy including radio-imaging, gene mutations, and expression levels of mRNAs and proteins [8].

Radiotherapy treatment of cancer patients induces a temporary cell cycle arrest that leads to either the cells repairing their DNA or, if the repair mechanism fails, cell death. The main type of DNA damage induced by ionizing radiation (IR) is double strand breaks (DSBs), the deadliest form of damage and the leading cause of cell death. In response to the induced DSBs, the cell activates a DNA damage response (DDR) involving both transducer and signal effector proteins including ATM, the MNR complex (MRE11, Rad50, and NBS1), and ATR [9]. The molecules involved in DNA DSBs signaling pathways are then excellent candidates for evaluation as radiosensitivity biomarkers since cells with a defective DNA damage response (DDR) have reduced ability to repair lethal radiation-induced DNA DSBs and are therefore more sensitive to irradiation, as seen in radiosensitivity cancer-prone syndromes such as ataxia telangiectasia [10] or Nijmegen breakage syndrome [11]. Previous studies have shown that, in a large series of malignant neoplasms, *ATM* somatic mutations are associated with tumor control following radiotherapy [12] and that the overexpression of MRN complex proteins in LARC treated with nCRT correlates with poor response and prognosis [13]; moreover, the mechanism of radioresistance through ATM pathway in LARC has been poorly explored [14,15,16,17,18].

In this study we aim to evaluate the association between somatic pathogenic variants in DDR genes, *ATM*, the *MNR* complex, and *ATR*, and response to radiotherapy in patients affected by LARC with either a pCR or pPR, using a NGS-dedicated panel specifically designed for FFPE samples to assess them as potential predictive factors of radiosensitivity.

## 2. Materials and Methods

### 2.1. Patients and Histological Evaluation

Thirteen patients with LARC TRG0, i.e., pCR, and 13 patients with LARC TRG 1–3, i.e., pPR, diagnosed at Sant’Andrea University Hospital of Rome between 2009 and 2018 were selected from histopathology archives.

The inclusion criteria were that all diagnostic and therapeutic procedures (from biopsy to TME) were performed at Sant’Andrea University Hospital.

All of the patients were diagnosed with adenocarcinoma on endoscopic biopsy and underwent clinical staging through nuclear magnetic resonance (NMR) for local disease and abdomen and chest computed tomography (CT) for systemic disease; according to multidisciplinary team decision meetings, all of the patients received 5-fluorouracil or capecitabine (Xeloda) along with a dose of 45–50 Gy radiation administrated over 5 weeks and underwent TME or abdominal-perineal resection after 6–10 weeks from the end of therapy.

Surgical samples were macroscopically and microscopically evaluated by experienced gastrointestinal pathologists according to guidelines’ consensus [19]. In brief, fresh surgical samples were evaluated for the integrity of the mesorectal plane and inked on circumferential margin. After fixation the entire area of irradiated neoplasia was sampled for microscopical evaluation as well as all of the lymph nodes. The macroscopic and microscopic distance between the neoplasia and circumferential margin was reported.

Upon histological examination the tumor regression grade (TRG) was scored according to the American Joint Committee on Cancer [20] grading system.

For the purpose of this study, the slides of endoscopic biopsy and surgical samples were revised to confirm the diagnosis and TRG grade by an experienced gastrointestinal pathologist (EP). All of the patients signed a written consent form, approved by the ethical committee, at the time of the rectal biopsy for research purposes.

### 2.2. DNA Isolation

A section of 5 to 10 μm of formalin-fixed paraffin-embedded (FFPE) pre-treatment biopsy was microdissected under microscope to ensure that each neoplastic sample contained at least 70% of tumoral cells tissue avoiding normal mucosa and necrosis.

Briefly, total DNA was extracted using GeneRead FFPE DNA Kit (QIAGEN, Hilden, Germany), according to manufacturer’s instructions, with pre-treatment of DNA FFPE samples with uracil DNA glycosylase (UDG) to minimize sequence artefacts due to cytosine deamination.

DNA was quantified using the Nanodrop System (Thermo Fisher Scientific, Waltaham, MA, USA) and Qubit dsDNA HS assay (Life Technologies, Waltaham, MA, USA).

### 2.3. NGS Analysis

NGS sequencing was performed using Ion PGM™ platform (Thermo Fisher Scientific, Carlsbad, CA, USA) on Ion 318™ Chip v2 BC support according to the manufacturer’s protocol. A custom Ion Ampliseq™ Panel (Thermo Fisher Scientific) containing 492 primer pairs in two pools, covering the exons and exon-intron boundaries of *ATM*, *ATR*, *MRE11*, *NBN*, and *RAD50* genes was used (Appendix A).

The prepared libraries’ quality was assessed using Qubit dsDNA HS assay and, after NGS sequencing, data analysis was performed using Torrent Suite version 5.12 and Ion Reporter Server System version 5.16.0.2 (Thermo Fisher Scientific, Carlsbad, CA, USA). Visual data analysis was performed using Integrative Genomics Viewer (IGV, https://igv.org/, version 2.3, Broad Institute and the Regents of the University of California, CA, USA). The amplification and sequencing conditions are described in Appendix A.

### 2.4. Somatic Variants Analysis and Classification

For sequencing data analysis, we used Ion Reporter Server System pipelines that include the following steps: read alignment, variant calling, variant annotation and reporting, and generation of QC matrices. Full details concerning the analysis filters applied are available in Appendix A.

The identified variants were filtered by consulting the databases (accessed on 5 July 2022) COSMIC (https://cancer.sanger.ac.uk/cosmic), ClinVar (https://www.ncbi.nlm.nih.gov/clinvar/), GnomAD (https://gnomad.broadinstitute.org/, and dbSNP (https://www.ncbi.nlm.nih.gov/snp/). Loss-of-function variants, frameshift variants, nucleotide insertions/deletions, gain/loss of stop codons, and splice site alterations were included. To evaluate the impact of missense variants on the structure and function of proteins, we consulted the prediction tools SIFT (https://sift.bii.a-star.edu.sg), FATHMM (http://fathmm.biocompute.org.uk), Mutation Taster (http://www.mutationtaster.org), Provean (http://provean.jcvi.org), and PolyPhen2 (http://genetics.bwh.harvard.edu/pph2). Variants were included when predicted as possibly pathogenic in at least three of five prediction programs used and/or scored with a CADD value > 20 (http://cadd.gs.washington.edu) or highlighted as ‘Driver’ in Cancer Genome Interpreter database (https://www.cancergenomeinterpreter.org/). Finally, we used the annotation tool and search engine Varsome (https://varsome.com/) which allows for the classification of the variants criteria according to the American the College of Medical Genetics and Genomics (ACMG) criteria [21].

Variants were reported using the Human Genome Variation Society (HGVS) guidelines (https://varnomen.hgvs.org/) nomenclature.

### 2.5. Single Nucleotide Variant Validation with ddPCR and Sanger Sequencing

To validate pathogenic variants identified with a VAF < 10%, digital droplets PCR assay (ddPCR) was performed on QX200 ddPCR System (Biorad, Hercules, CA, USA), using the nonspecific DNA binding properties of EvaGreen (EG) dye. Each 22 µL ddPCR of master mix was partitioned and plated using QX200 droplet generation according to the manufacturer’s instructions. PCR primers were designed to validate the pathogenic variant identified in the cancer tissue sample. Full details of methods concerning primer design and amplification conditions are provided in the Appendix A.

To discriminate germline mutations, bidirectional Sanger sequencing of selected variants was performed on matched constitutional DNA using the BigDye Terminator, v3.1, Cycle Sequencing Kit, and an SeqStudio™ Genetic Analyzer System Analyzer (Thermofisher Scientific, USA). Primers are available upon request.

### 2.6. Statistical Analysis

A comparison of demographic and clinical variables between the groups was performed with unpaired the Mann–Whitney U test for continuous variables and χ^2^ test (or Fisher’s exact test, if appropriate) for categorical data, respectively. The Spearman correlation coefficient was used to assess the relationship between two variables. The *p*-values lower than or equal to 0.05 were considered statistically significant.

## 3. Results

### 3.1. Patients

A total of 26 patients, 13 pCR and 13 pPR, were included in this study (Figure 1). The clinical–pathological data are listed in Table 1. This cohort included 17 males (65.4%) and 9 females (34.6%). The median age at diagnosis was 65.7 years (range: 30–77 years). Two (7.7%) patients were clinically staged as I before treatment, 9 (34.6%) and 15 (57.7%) patients were clinically staged as II and III, respectively. In the pPR group, two patients were classified as TRG1, and eight and three patients were classified as TRG2 and TRG3, respectively. Local or distant recurrences occurred in 9/26 (34.6%) patients, and this was significantly observed in patients with pPR (*p* = 0.0112).

### 3.2. Mutational Analysis

A total of 24 samples, 13 pCR and 11 pPR, were subjected to targeted sequencing with a panel of genes involved in DNA repair by HR: ATM, the MRN genes complex, and ATR gene. Two cases were excluded for poor quality of DNA. PGM sequencing produced an average of 604.397 mapped reads per patient, with the mean read length being 95 bp. The average read depth per sample was 1700 reads, with a mean (percentage of reads on target of 95.4%. The mean percentage of regions of interest (ROI), covered at least by 500×, was 98.66% with uniformity by 96.07%.

Among all of the patients in our cohort, 349 unique variants with an average of fifteen mutations per case in both groups were identified. According to the ACMG criteria, 341 (97.7%) are classified as benign, 7 (2%) are classified as VUS, and 1 (0.3%) has been classified as pathogenetic (Table 2).

In particular, a pathogenetic variant was identified in the RAD50 gene in a patient with pCR. It represented a frameshift mutation null variant c.3271G>T; p.(Glu1091*) in exon 21 with a VAF of 8%, confirmed by ddPCR (Appendix A). In the same patient, an intronic c.3171 + 7T>G variant on the ATR gene was identified. One patient (n.9) in the pCR group harbored an ATM gene missense germline mutation (c5890A>G) in the FAT/kinase domain; patient 2 carried two missense somatic variants in RAD50 gene and 1 missense somatic variant in MRE11. The two remaining non-benign missense variants were found in 2 (18%) out of 11 patients in the pPR group: in patient 16, a germline variant in the NBN gene and in patient 24, a somatic variant in the pincer domain of the ATM gene.

### 3.3. Correlation of Genomic Alterations with TCGA Data

We next examined The Cancer Genome Atlas (TCGA) data to identify the frequency of mutations in five genes involved in the DDR pathway across the specific clinicopathological setting of LARC.

Firstly, we selected samples in the TCGA Rectum Adenocarcinoma (READ) database filtered as: organ of origin as “colon” and treatment type as “radiation therapy” and disease type as “adenomas and adenocarcinomas” and primary site as “rectum” and sample type as “primary tumor” and Vep impact as “high”.

Comparison with the TCGA cohort (n = 78) shows broadly similar frequencies of mutations in DDR genes in our smaller cohort. The frequencies of mutations from TCGA data and our internal dataset are depicted in Figure 2.

## 4. Discussion

In locally advanced rectal cancer (LARC), an adjuvant regimen of preoperative neoadjuvant chemoradiotherapy (nCRT) followed by surgery is the standard of care since it has been proved to reduce the exceptionally high rate of local and distant recurrences; however, the effect on tumor regression by chemoradiotherapy in each single patient is unpredictable.

Ionizing radiation treatment in cancer therapy is known to induce DSBs (double-strand breaks). At the DSBs sites, the MRN complex recruits the ATM kinase and the direct interaction with this complex induces its activation by auto-phosphorylation and monomerization.

To our knowledge, this is the first study that investigates mutations in *ATR*, *ATM*, and *MRE11-RAD50-NBN* genes in LARC using a NGS-dedicated panel specifically designed for FFPE samples as most studies have in fact evaluated gene mutations between the two groups using large cancer panels that are commercially available in which *ATM* sequences are usually present and *RAD50* sequences are seldom present [15,22,23].

According to bioinformatic tools, excluding those classified as benign/probably benign, we identified eight variants of which there were seven of unknown clinical significance classified as VUS in the *ATM* (2), *RAD50* (2), *ATR* (1) *Mre11* (1), and *NBN* (1) genes and one in the *RAD50* gene classified as a pathogenetic variant.

The truncating variant in the *RAD50* gene c.3271G>T (p.Glu1091*), reported also in Clinvar as pathogenic, was identified in patient p3 of the pCR group, i.e., in a patient who underwent complete tumor regression (TRG0); the same patient harbored the intronic variant c.3171 + 7G>T in the *ATR* gene, which is not described in the literature and is classified as VUS in Varsome. Loss of function variants in the *RAD50* gene are known to be pathogenic; in fact, cells from a patient compound heterozygous for *RAD50* pathogenic variants show chromosomal instability, radiosensitivity, and fail to form MRN foci induced by DNA damage [24].

As a component of the MRN complex, RAD50 is involved in DNA repair following ionizing radiation. Chen et al. [25] observed that RAD50 knockdown in CRC cell lines sensitized cells to irradiation effects thereby reducing DSB repair efficiency. On this basis, it has been suggested that a tumor with a deficiency in MRN complexes could be more sensitive to the DNA-damaging effect of radiotherapy [13]. We did not investigate the expression of RAD50 at the protein or mRNA level in our samples; however, since we have identified a protein truncation gene mutation, it is possible to speculate that it could cause a reduction at the protein level and the MRN complex and, therefore, it is impaired in that sample DNA damage repair.

For the two variants in the *RAD50* gene identified in pt. two of the pCR group and classified as VUS in Varsome, it is difficult to speculate a role. The incidence of *RAD50* mutations in our series is in agreement with TCGA data and previous reports. Zhou et al. [26] reported 3% of *RAD50* non-silent mutations in a large series of 406 Chinese samples of CRC; however, the fraction of rectal cancer was not specified and the relationship with radiotherapy treatment was not investigated [27].

The *ATM* gene is a master controller for cell response to DNA damage. It is recruited upon DNA double-strand breaks and is involved in DNA repair. In vitro studies have shown increased radiosensitivity after pharmacologic inhibition of *ATM* [28,29] and in cancer harboring *ATM* mutations [12].

Two out of seven variants classified as VUS by Varsome were identified in the *ATM* gene and in particular, one in patient p9 (c.5890A>G) of the pCR group and one in patient p24 (c.4414T>G) of the pPR group. The former is a missense variant that causes the non-conservative substitution of lysine with glutamate in position 1964 of the FAT domain of the ATM protein. The variant with VAF 49.78% was confirmed to be germinal, has a frequency in the general population of 9.9 × 10^−5^, and has been reported in individuals with hereditary breast cancer [30,31] and Lynch syndrome [32]. The latter is a somatic missense variant in the pincer domain that causes substitution of leucine with valine at codon 147 (p.Leu1472Val). This variant has been reported previously in patients with breast, ovarian, melanoma, and prostate cancer [33,34,35,36].

It can be inferred that the position of these variants may have affected the response of rectal cancer to radioactivity since the FAT/Kinase domain is responsible for phosphorylate activity; however, since they are VUS, more data are needed to evaluate if the variants located in the FAT domain may confer different sensibility to radiation [37].

The role of *ATM* in LARC’s response to radiotherapy has been explored either at the protein expression level or in gene mutations. It has been reported that high expression of *ATM*, especially when combined with MRE11, is associated with worse DFS in rectal cancer treated with neoadjuvant radiotherapy [38].

Previous studies that have investigated the *ATM* gene in LARC as part of a larger NGS panel have reported controversial results on a possible relationship between gene mutations and radiotherapy response. Matos do Canto et al. [15] found 2 out of 33 LARC patients with *ATM* gene variants, both in the subgroup of pathological incomplete response after neoadjuvant chemoradiotherapy. Toomey et al. [22] in a series of 31 LARC patients reported a variant of *ATM* in one out of five patients with a complete pathological response.

A higher rate was reported by Douglas et al. [23] who found *ATM* gene variants in 10 out of 17 LARC cases without any significant difference between TRG0 and TRG2-3 patients.

Taken together, these results do not support a predictive role of *ATM* in selecting patients’ sensitivity to radiotherapy in rectal cancer as shown in other neoplasms [12].

In patient p16 of the pPR group, we identified a missense germline mutation c.511A>G, p.(Ile171Val) in the *NBN* gene classified according to the ACMG criteria as VUS. Some germline *NBN* variants have been associated with an increased risk of developing breast cancer; however, no data are available for colon cancer. Recently, a prognostic role for *NBN* somatic mutations has been suggested in a subset of colorectal cancer although in the study, no rectal cancer was included [39].

We recognize that the strength of our study, i.e., the meticulous case selection, represents at the same time its limit since the number of samples investigated is relatively small.

## 5. Conclusions

Our data show that the genes involved in the HR pathway are rarely mutated in LARC, thus, their role as predictive biomarkers to nCRT response seems limited; however, given the identification of a missense mutation in RAD50 in one case of pCR, it could be worth exploring its potential role as a biomarker in larger series.

## Figures and Tables

**Figure 1 biomedicines-10-03247-f001:**
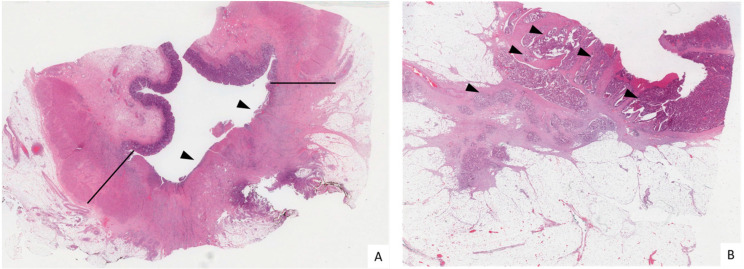
Histological sections of two surgical samples of LARC after nCRT. (**A**) TRG0 sample. In the area of the neoplasm, included between the black lines, there is extensive fibrosis of the rectal wall without residual cancer. The mucosa is ulcerated (arrow heads). (**B**) TRG3 sample. The rectal wall is still massively infiltrated by neoplastic glands (arrow heads). (Hematoxylin/Eosin, 5×).

**Figure 2 biomedicines-10-03247-f002:**
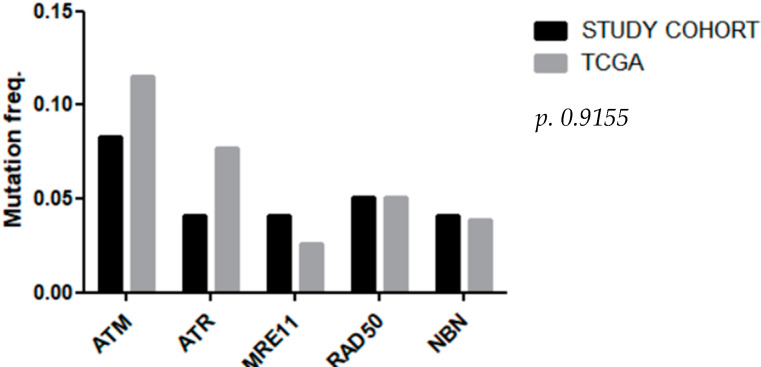
Mutation frequencies of *ATM*, *ATR*, *MRE11*, *RAD50*, and *NBN* in TCGA data and the study cohort.

**Table 1 biomedicines-10-03247-t001:** Clinical–pathological characteristics of the 26 patients included in the study.

Characteristics		Total pts (N = 26)	pCR (N = 13)	pPR (N = 13)	*p* Value
**Age at diagnosis**		65.7 ± 10.9	67.1 ± 2.4	64.5 ± 3.6	*0.2773*
**Gender**	Male	17	8	9	*0.6802*
	Female	9	5	4	
**cT at diagnosis**	cT2	5	4	1	*0.2167*
	cT3	20	8	12	
	cT4	1	1	0	
**cN at diagnosis**	cN0	11	4	7	
	cN1	13	7	6	*0.2351*
	cN2	2	2	0	
**Surgery**	LAR	21	11	10	*0.5104*
	APR	5	2	3	
**ypT**	0	13	13	0	*0.5385*
	1	1	0	1	
	2	4	0	4	
	3	8	0	8	
**ypN**	0	22	13	9	
	1	2	0	2	*0.0941*
	2	2	0	2	
**Recurrence**	No	17	12	5	** *0.0112* **
	Yes	9	1	8	
**TRG**	0		13	0	*0.9248*
	1		0	2	
	2		0	8	
	3		0	3	

Abbrevations: pCR, pathological complete response; pPR, pathological partial response; cT, clinical tumor stage; cN, clinical nodal stage; ypT, post-neoadjuvant treatment pathological tumor stage, ypN post-neoadjuvant treatment pathological nodal stage; LAR, low anterior resection; APR, abdominoperineal resection; TRG, tumor regression grade.

**Table 2 biomedicines-10-03247-t002:** Class 1–3 variants identified in the study in FFPE tumor samples.

ID Sample	GENE	Genomic coordinate	HGVS cDNA	HGV protein	dbSNP	Type of Variants	S/G	VAF (%)	CLINVAR	VARSOME	SIFT	POLYPHEN	FATHMM	PROVEAN	GnomAD	COSMIC ID	CADD	MUTATION TASTER	CGI
											(1–0)	(0–1)	(Score) (0–1)	(Cutoff −2.5)	(f)				
**p9**	*ATM*	chr11:108310287A>G	c.5890A>G	p.(Lys1964Glu)	rs201963507	missense	G	49.78	Confl. Int. of pathogenecity	VUS	0.18	0.009	Pathogenic	N (−0.63)	0.0000996	2110551	22.2	DC	P
**pCR**													(0.99)						
**p3**	*ATR*	chr3:142549472C>A	c.3171+7G>T	-	N/A	intronic	G	38.40	N/A	VUS	N/A	N/A	N/A	N/A	N/A	N/A	8.9	N/A	P
**pCR**	*RAD50*	chr5:132618176G>T	c.3271G>T	p.(Glu1091*)	N/A	frameshift	S	7.99	Pathogenic	Pathogenic	N/A	N/A	Pathogenic	N/A	N/A	N/A	42.0	DC	D
													(0.9889)						
**p2**	*RAD50*	chr5:132557450G>T	c.126G>T	p.(Lys42Asn)	rs754823399	missense	S	20.11	N/A	VUS	0	1.0	Pathogenic	DE (−4.647)	N/A	N/A	28.2	DC	D
**pCR**													(0.7048)						
	*RAD50*	chr5:132587587A>G	c.782A>C	p.(Asn261Ser)	N/A	missense	S	21.96	N/A	VUS	0.49	0.003	Pathogenic	N (−0.364)	N/A	N/A	22.0	DC	P
													(0.9557)						
	*MRE11*	chr11:94490905C>A	c.81G>T	p.(Glu27Asp)	rs190031653	missense	S	22.88	N/A	VUS	0.004	0.922	Pathogenic	DE (−2.71)	N/A	N/A	23.4	DC	D
													(0.9386)						
**p16**	*NBN*	chr8:89978293T>C	c.511A>G	p.(Ile171Val)	rs61754966	missense	G	63.30	Confl. Int. of pathogenecity	VUS	0.02	0.994	Pathogenic	N (−0.8)	0.0012	9496534	23.3	DC	P
**pPR**													(0.9318)						
**p24**	*ATM*	chr11:108289779T>G	c.4414T>G	p.(Leu1472Val)	rs539676759	missense	S	31.10	Confl. Int. of pathogenecity	VUS	0	0.858	Pathogenic	N (−2.33)	0.0000854	9494445	21.8	DC	P
**pPR**													(0.8111)						

Abbreviations: pCR, pathological complete responder; pPR, pathological partial responder; HGVS, Human Genome Variation Society; dbSNP, Single Nucleotide Polymorphism Database; rs, reference SNP; S, somatic variant; G, germinal variant; VAF, variant allele fraction; VUS, variant of uncertain significance; DE, deleterious; N, neutral; DC, disease causing; P, passenger; D, driver; N/A, data not available.

## Data Availability

Not applicable.

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
