# Peer review of "Somatic NGS Analysis of DNA Damage Response (DDR) Genes ATM, MRE11A, RAD50, NBN, and ATR in Locally Advanced Rectal Cancer Treated with Neoadjuvant Chemo-Radiotherapy"

_biomedicines, 2022, doi:10.3390/biomedicines10123247_

Round 1
Reviewer 1 Report
In this manuscript, the authors assessed the host DDR genes in rectal cancer patients receiving neoadjuvant chemio-radiotherapy. The authors focused on the following genes ATM, MRE11A, RAD 50, NBN and ATR. The authors claimed that missense mutation in RAD50 could be a possible predictive biomarker for nCRT in LARC.
I have major concerns on the manuscript in the present form
1- The number of patients is relatively small to get a conclusion
2- Why the authors focused on these specific genes. What are rationales other than they are DDR genes? They are a lot of DDR genes, so why focusing on these ones
3- I wonder why the authors did not screen several DDR genes , especially they used NGS, and from the data they can go for the resulted ones.
4- Conclusion is not solid at all, to have one patient with missense mutations in the RD50 and then the authors concluded to use RD50 as a biomarker????
Reviewer 2 Report
The manuscript was aimed to evaluate the association between somatic mutations in DNA damage repair (DDR) pathways (e.g. ATM, ATR and the components of MRN complex) and radiosensitivity of the colorectal cancer (CRC).
26 clinical samples were analyzed. It was shown that 13 patients reached a complete response (pCR) to radiotherapy , whereas the remaining 13 patients demonstrated partial or no response (pPR).
The major finding of this study is a frameshit mutation in exon 21 of RAD50 gene, a well-known component of MRN complex involved in repair of the DNA double-stand breaks (DSBs) induced by ionizing radiation. This was observed in tumor specimen from pCR group. This data is in a proper fit with a general concept illustrating low sensitivity to DNA-damaging agents (radio- and chemotherapy) in cancer cells exhibiting functional HR and NHEJ-pathways.
Given that homology-mediated DNA repair pathway plays a crucial role in DSBs repair, I also recommend the authors to perform the mutational analysis of Rad51 recombinase (as optional)
Round 2
Reviewer 1 Report
The authors replied to most of my concerns and the manuscript is modified significantly.
One main point, I would suggest to confirm NSG results with qPCR for the selected genes for more confirmation, if still RNAs are present, or if the authors have FFPE, then they can try protein expression of these targets by IHC or IF.
Author Response
Answer to Reviewer 1
Point 1.
“One main point, I would suggest to confirm NSG results with qPCR for the selected genes for more confirmation, if still RNAs are present, or if the authors have FFPE, then they can try protein expression of these targets by IHC or IF”.
Answer 1.
We thank the reviewer for the comment and new suggestion.
As reported in “Methods” section, we confirmed the variants identified with NGS through Sanger sequencing (variants with allele frequency >10%) or ddPCR (i.e RAD50 frameshift mutation that showed a VAF<10%).
Moreover, to discriminate germline mutations we performed Sanger sequencing on matched non neoplastic DNA.
The aim of our project was mutational analysis of the coding sequence and exon-intron boundaries of the 5 genes selected involved in DDR pathway. As suggested by the reviewer the investigation of the expression of gene products would be very interesting. However, due to the small size of bioptic samples and the poor quality of RNA extracted from FFPE specimens, we couldn’t pursue this objective in our study.
We are planning to collect prospectively new samples of LARC to perform gene expression study at RNA and protein level taking into account the data obtained from mutational analysis.
Round 3
Reviewer 1 Report
no further comments